# Integration of Response Surface Methodology (RSM) and Principal Component Analysis (PCA) as an Optimization Tool for Polymer Inclusion Membrane Based-Optodes Designed for Hg(II), Cd(II), and Pb(II)

**DOI:** 10.3390/membranes11040288

**Published:** 2021-04-14

**Authors:** Jeniffer García-Beleño, Eduardo Rodríguez de San Miguel

**Affiliations:** Departamento de Química Analítica, Facultad de Química, Universidad Nacional Autónoma de Mexico (UNAM), Ciudad Universitaria, 04510 Ciudad de Mexico, Mexico; jgarciab@comunidad.unam.mx

**Keywords:** polymer inclusion membrane, optode, desirability function, parameters optimization, response surface methodology, principal component analysis

## Abstract

An optimization of the composition of polymer inclusion membrane (PIM)-based optodes, and their exposure times to metal ion solutions (Hg(II), Cd(II), and Pb(II)) was performed using two different chromophores, diphenylthiocarbazone (dithizone) and 1-(2-pyridylazo)-2-naphthol (PAN). Four factors were evaluated (chromophore (0.06–1 mg), cellulose triacetate (25–100 mg) and plasticizer amounts (25–100 mg), and exposure time (20–80 min)). Derringer’s desirability functions values were employed as response variables to perform the optimization obtained from the results of three different processes of spectral data treatment: two full-spectrum methods (M1 and M3) and one band-based method (M2). The three different methods were compared using a heatmap of the coefficients and dendrograms of the Principal Component Analysis (PCA)reductions of their desirability functions. The final recommended M3 processing method, i.e., using the scores values of the first two principal components in PCA after subtraction of the normalized spectra of the membranes before and after complexation, gave more discernable differences between the PIMs in the Design of Experiments (DoE), as the nodes among samples appeared at longer distances and varyingly distributed in the dendrogram analysis. The optimal values were time of 35–65 min, 0.53 mg–1.0 mg of chromophores, plasticizers 34.4–71.9 of chromophores, and 62.5–100 mg of CTA, depending on the metal ion. In addition, the method yielded the best outcomes in terms of interpretability and an easily discernable color change so that it is recommended as a novel optimization method for this kind of PIM optode.

## 1. Introduction

Chemists often work in very complex systems, where the estimation of the variables and their interactions affecting a phenomenon or finding the differences between two or more extremely complex sample classes may not be immediately apparent from a superficial evaluation. Subsequently, maximizing the information gained from a chemical experiment becomes a crucial step to reduce the time and cost necessary for studying chemical systems. Chemometrics is a prominent area dedicated to developing multivariate strategies for chemical data evaluation and interpretation [1]. Over the years, chemometrics has become an important chemical discipline, including the incorporation of significant improvements in design and selection of optimal experimental procedures, and advanced methods for analysis of chemical data [2]. With a suitable design of experiments (DoE) as a basis for experimentation, optimal information about an investigated system can be gained and the cost of gaining that information can be minimized. In chemistry and related fields, DoE has been used in the optimization of organic synthesis, peptide design, cheese manufacture, bread-making, investigation of calibration process parameters, and various other applications [3]. With the continuous technological progress of instrumental techniques for analytical purposes, multiple responses are now easily generated in the study of chemical systems, e.g., NIR, NMR, Raman, MS, and UV-VIS spectra. Multivariate analysis (MVA) applied to this type of chemical responses is mandatory to use all the information contained within the spectra, in the analysis and interpretation of the data. By coupling both chemometric areas, DoE and MVA, a valuable way to study complex chemical systems is created. In this context, such connection has been used, either in a complementary or integral form, in diverse areas, such as product development in a continuous process [4], the development of a drug product [5], size exclusion chromatography for development of silica-based stationary phases [6], for undertaking metabolomic studies [7], for enhancing the performance of cathodes [8], for studying the cadmium biosorption process [9], for studying the effects of physical properties of dosage forms [10], to determine the moisture content in mAb lyophilisates [11], and to create solvent maps to identify safer alternatives to toxic/hazardous solvents, and also in the optimization of an SNAr reaction [12], among others. Despite such a wide range of applications, in the area of optodes for metal ions, although DoE strategies have been employed [13,14], little advantage has been taken from the simultaneous coupling between DoE and MVA concerning the manufacturing and optimization of sensor composition. This article aims to provide evidence of the utility of this approach by integration of the response surface methodology, RSM, an area of DoE, and principal component analysis, PCA, an area of MVA, for optimizing the composition and exposure time of polymer inclusion membrane (PIM) based-optodes for the measurement of metal ions. Along the work, a Doehlert design matrix [15] coupled to the Derringer’s desirability function (DF, [16]) is described and analyzed using an algebraic transformation of spectral data before PCA analyses of the responses allows an easy and integral form to optimize PIM optodes for sensing three metals (Hg(II), Cd(II), and Pb(II)) with two chromophore agents (dithizone and PAN). PIMs are a type of membrane in which the chromophore agent is contained within the polymeric network of a non-porous support in the presence or absence of a plasticizer. They have been used for sensing Co(II) [17,18], Al(III) [14,19,20], Zn(II) [21], and Cu(II) [22], among other metal ions. A recent overview of PIM applications, including sensors, has been presented elsewhere [23].

## 2. Materials and Methods

### 2.1. Reagents 

Cellulose triacetate (CTA, Aldrich. Darmstadt, Germany), 2-nitrophenyl octyl ether (2NPOE 99%, Aldrich), tris(2-ethylhexyl) phosphate (THEP 97%, Aldrich), 1-(2-Pyridylazo)-2-naphthol (PAN indicator grade, Aldrich), diphenylthiocarbazone (Dithizone A.C.S reagent, Aldrich), dichloromethane (99.99%, J.T. Baker, PA, USA), and ethanol (Analyka, 99.9%) were employed in PIMs preparation. The solutions of metal ions were prepared at a concentration of 2 × 10^−5^ mol/L employing the following metallic salts: lead nitrate Pb(NO_3_)_2_ (A.C.S reagent 99.5%, Fermont), cadmium nitrate Cd(NO_3_)_2_·4H_2_O (J. T. Baker 99.1%), and mercury nitrate monohydrate Hg(NO_3_)_2_·H_2_O (A.C.S reagent 99.5%, Fermont). A solution of 2-N-morpholino) ethanesulphonic acid (MES hydrate, >99.5%, Sigma-Aldrich) 10^−2^ mol L^−1^ was employed for buffering the systems.

### 2.2. Instruments

A Perkin Elmer model Lambda 2 UV/vis spectrophotometer was used to record all absorbance spectra (400 nm–800 nm). The pH of metal solutions was measured using a SevenCompact pH meter S220 with a combined glass electrode Cole-Parmer 62014, Mettler Toledo, Ciudad de México, México. A Burrel 75 mechanical shaker was employed to shake the metal solutions containing the membranes.

### 2.3. Doehlert Experimental Design

To find the optimal composition of the optode, it is necessary to evaluate the amounts of components used, as well as the agitation time to which the sensor is exposed to the aqueous solution containing the metal. In this way, four factors were evaluate (CTA, plasticizer, and chromophore contents and time), each one at different levels. This led to the Doehlert design matrix being chosen [15], with 21 experiments as shown in Table 1, where the real and coded levels of the variables are indicated. These levels were chosen based on preliminary results, where PIMs were prepared employing different compositions and observations were made with respect to the possibility of membrane formation and its resistance. It is important to mention that along the work, all statistical analyses were performed using coded values of the variables, such that they were not dependent on the measured scale range. As observed, six experimental designs were executed for the individual determination of the three ions (Hg(II), Cd(II), and Pb(II)) in solution: three using dithizone and three using PAN as chromophores.

#### 2.3.1. Membrane Preparation

PIMs were prepared according to the procedure described elsewhere [24]. Briefly, weighted amounts of CTA, plasticizer (THEP or 2NPOE), and chromophore (PAN or Dithizone, Dz) (see Table 1) were dissolved in a 1:9 (*v*/*v*) dichloromethane-ethanol mixture. THEP was used for PIMs with PAN while 2NPOE for those with Dz, as good solubility behavior was observed using such combinations. The addition of ethanol during casting ensured PIM homogeneity. The mixture was stirred for 1 h until homogeneity, then poured into a Petri dish(Internal diameter of 5 cm) and allowed to evaporate for 24 h. Then, the membranes were detached from the Petri dish, adding water, and, subsequently, the visible spectrum was acquired.

#### 2.3.2. Measurements

The membranes were shaken in the presence of 30 mL of a solution containing the metal ion, at a concentration of $2\times 10^{-5}$ mol/L and pH 6.5, adjusted with 10^−2^ mol/L MES buffer. Each sensor was subjected to stirring at different times, as indicated in the design matrix (see Table 1). Once the agitation was finished, the membrane was dried at ambient temperature and the visible spectrum was measured to, subsequently, perform the data processing and analysis.

### 2.4. Optimization

#### 2.4.1. Response Surface Methodology (RSM)

The response surface methodology is a collection of mathematical and statistical techniques based on fitting a polynomial equation to a set of experimental data, in order to make statistical predictions [25]. This approach has been applied in investigations that involve analysis of the interaction of independent variables and their influence on an answer (dependent variables) [26,27]. In the area of PIMs, RMS has been used as a tool to establish the influence of the components of a membrane or pH and concentrations of feed or strip solutions in the determination of different metals [18,28]. 

#### 2.4.2. Principal Component Analysis (PCA)

This is a data reduction technique that aims to find new variables (principal components) that are linear functions of those in the original dataset, which successively maximize variance and are uncorrelated with each other, preserving as much variability as possible in the data but eliminating noise. PCA as a descriptive tool needs no distributional assumptions and, as such, it is very much an adaptive exploratory method, which can be used on numerical data of various types [29]. The central point is to reduce the original (*m*,*n*) data matrix *X* with *m* features (variables) and *n* objects (samples) to the following components parts that are linearly related according to the Equation (1): (1)X=A·F+E
where *A* are factor loadings and *F* factor scores. So, the linear combination of the loadings and scores that constituted the principal components can reproduce the original data matriz *X* with a minimal loss of information represented by the matrix of residuals *E*. In this manner, PCA provides a projection from the high-dimensional feature space on to a space defined by a few factors; they can also be used as a method for graphical representation of multidimensional data [30]. In the present case, the *X* matrix in Equation (1) is constituted by the combination of VIS spectra (columns) of all PIMs prepared along the 21 experimental runs (rows) for a particular chromophore/metal system, while *A* and *F* represent loadings and score matrices, respectively, which define the different principal components used in trying to reduce the number of variables to analyze. 

#### 2.4.3. Derringer’s Desirability Function

Before DoE analyses, the individual responses, i.e., PCs or absorbances (Y_i_), were first fitted to a polynomial function to give expected values of the responses; then, each estimated response variable was transformed to a desirability value d_i_, where 0 ≤ d_i_ ≤ 1 (Equation (2)):(2)$$d_{i}\left(Y_{i}\right)=\{\begin{array}{c} 0\\ {\left(\frac{Y_{i}-\mathrm{LSL}\;}{\mathrm{USL}-\mathrm{LSL}}\right)}^{S},\\ 1.0 \end{array}\begin{array}{c} Y_{i}<\mathrm{LSL}\\ \mathrm{LSL}\leq Y_{i}\leq \mathrm{USL}\\ Y_{i}>\mathrm{USL} \end{array}$$
where “LSL” and “USL” are the lower and upper specifications limits of the associated response Y_i_. The weight exponent “s” specifies the form of the response within the range of interest [26]. The value of d_i_ increases as the desirability of the corresponding response increases for an optimization case in which the estimated response has to be maximized; however, d_i_ varies if the response is required to be minimized according to Equation (3) [16,31]:(3)$$d_{i}\left(Y_{i}\right)=\{\begin{array}{c} 1\;\;\;\;\;\;\;\;\;\;\;\;\;\;\;\;\;\;\;\;\;\;\;Y_{i}<\mathrm{LSL}\\ {\left(\frac{Y_{i}-\mathrm{USL}}{\mathrm{USL}-\mathrm{LSL}}\right)}^{s},\;\;\;\;\mathrm{LSL}\leq Y_{i}\leq \mathrm{USL}\\ 0\;\;\;\;\;\;\;\;\;\;\;\;\;\;\;\;\;\;\;Y_{i}>\mathrm{USL} \end{array}$$

The individual desirabilities, d_i_, were then combined using a modified geometric mean as:(4)D=(d1I1·d2I2·…·dkIk)1∑Ik

This single value of D gives the overall assessment of the desirability of the combined response levels. DF is one of the most used methods for optimization of multiple response processes in science and engineering [32]. By considering all responses in measurement through a weighted geometric mean, it provides the possibility of predicting the optimal levels for the independent variables. The function varies between the value of zero, which suggests that the answer is completely unacceptable, to the value of 1, which means that the answer corresponds exactly to the target value. In Equation (4), I_k_ is the impact coefficient, ranging between 1 and 5 [33]. In the present case, the default value of 3 for all k was assumed, as the goal was to give the same importance to each d_i_.

#### 2.4.4. Heat Maps

A heatmap is a two-dimensional visual representation of data using colors, where the colors all represent different values. Heatmaps can provide an efficient and comprehensive overview of a topic at a glance and unlike charts or tables they are direct data visualization tools that are more self-explanatory and easy to read [34].

#### 2.4.5. Hierarchical Cluster Analysis (HCA)

Cluster analysis includes several powerful algorithms and methods for grouping objects of similar kinds into organized categories. It is an exploratory analysis tool that aims to sort different objects into groups in such a way that the degree of association between two objects is maximal if they belong to the same group and minimal if not [35]. The results are visually represented by a two-dimensional dendrogram, a tree diagram that lists each observation according to the similarities (distances) to the others. In hierarchic methods, all objects begin alone in groups of size one, and groups that are closer together are merged. One could use either the original X variables or PCA scores to determine the distance. The usage of PCA scores can provide collinearity and noise reduction benefits but requires the specification of the appropriate number of PCs. Additionally, given the input variables (X variables or PC scores), one can then choose either the Euclidean or the Mahalanobis distance to complete the definition of the distance measure. In the present case, the Mahalanobis distance was used as it accounts for dominant multivariate directions in the data when performing cluster analysis. In the analysis, the desirability values for the three metals at each processing method were reduced by PCA and used as input for the HCA methodology. 

### 2.5. Data Presentation

The optimization was made in three different modes; in the first one (M1), the PCA reduction of the spectra after complexation was performed and the scores values of two first principal components were employed as a response in the experimental design. The second mode (M2) was made by employing both the absorbances of the free chromophore and that of the formed complex as a response in the experimental design. The third mode (M3) was made by first normalizing the VIS spectrum to the highest absorbance value of all the spectra, before and after complexation, to preserve the quantitative relationships that might exist among the absorbance values and, subsequently, subtracting each pair of spectra. This data set was used to perform the PCA reduction and the score values of the two first principal components were used as response variables in the experimental design. VIS spectra were baseline corrected before analyses. Statgraphics Centurion 16 (Statgraphics Technologies, Inc., The Plains, VA USA), Unscrambler 10.5.1 (CAMO Analytics, Oslo, Norway) and PLS-Toolbox 8.7.1 (Eigenvector Research, Inc., Manson, WA, USA), and Plotly Chart Studio (Plotly, Montreal, QC, Canada) software were used for data processing and analysis. 

## 3. Results and Discussions

Since the reactions of the metal ions with the used chromophores form a colored complex, colorimetric detection of these ions can be carried out using polymeric inclusion membranes in which the chromophore is contained in the polymeric network of a non-porous support [23]. The behavior of the PIM used as the optode is determined by its composition, the range of concentrations in which it works, and the time during which the sensor is exposed to the analyte-containing solution [36]. Therefore, parameters, such as the quantities of the components of the membrane and the time of exposure to the solution containing the metal ion, were optimized. This was carried out using a design of experiments approach, which considers all the factors simultaneously to determine the influence of the factors and their interactions on the behavior of the optode. In this case, a four-factor Doehlert matrix was used, each at different levels, as described in the experimental part. One of the main challenges was to find an adequate response for the DoE matrix, in such a way that it preserves most information about the membrane/(membrane+metal) system. Therefore, the three ways (M1, M2, and M3) to evaluate the response and including it in the design of experiments to perform the optimization of the sensor’s composition were carried out and subsequently compared. Each mode was employed for sensors made with the three different metal ions (Cd(II), Hg(II), and Pb(II)) and two chromophore agents (Dithizone, Dz, and PAN).

### 3.1. PCA of the Spectra after Complexation (M1)

In Figure 1, an example of raw spectra data of the PIMs for the Dz-Hg(II) system is shown. As observed, very different VIS spectrum profiles are obtained depending on the membrane composition. Although the most notable change is the baseline shift, the appearance and disappearance of bands at about 500 and 625 nm and their shifts are observed as well. These bands are related to the chromophore and its metal complex. After dimensionality reduction, two PCs explain about 85% of the variability in the data (Figure 2), allowing them to be used as a response in the DoE, and evaluation of their effect in conjunction through the desirability function (Figure 3).

From the loadings graph of the first two principal components (Figure 3), which shows the relationship between these new variables and the original ones, it is observed that the region of the spectrum between 490 and 510 nm has a great influence on the first principal component PC-1 (blue line in Figure 3); this region of the spectrum corresponds to the formed Dz-Hg(II) complex.

Similarly, with respect to PC-2 (red line in Figure 3), the influence of the region of the spectrum corresponding to the uncomplexed dithizone (510 and 615 nm) is denoted. After comparison of both loadings, it is clear that an inverse behavior is observed in the zone around 430–560 nm, i.e., positive values for the loadings in PC-1 contrast with negative ones for those of PC-2, meaning that some contributions of this spectral region cancel out. On the contrary, the contributions of both PCs in the region above 560 nm are added together, as both have positive values, indicating that the analysis is strongly affected by the VIS spectral region of the uncomplexed form of the chromophore. This means that the analysis is practically performed based on the reduction in intensity of the band of Dz due to complexation with the metal ion. After the score values of both PCs were considered as responses in the experimental design for maximization, the fitted equation for their desirability was computed as:D = 0.576507 − 0.09592 × Dz − 0.235797 × Time × CTA (5)

The significant terms (95% confidence interval), Dz and Time × CTA, were chosen according to the Pareto (Figure 4) and ANOVA (Table 2) analyses. With the response surface of the desirability function, an optimal membrane was then selected (Table 3).

The negative influence on both significant factors indicates that an increase in, for example, the amount of Dz decreases the desirability function. Considering that the desirability function jointly represents the PC-1 and PC-2 responses and these, in turn, are related to the behavior of the membrane before and after complexation, an increase in the amount of dithizone can chemically be interpreted as the greater the amount of chromophore, the more saturated the membrane becomes.

Therefore, due to the low metal concentration used, the band of the free ligand mask existed with that of the metal complex. Now, regarding the Time*CTA interaction, this means that the time factor affects the response at each CTA level differently. For example, thick membranes with high CTA content must be exposed to short periods of time to obtain a better performance. 

Similar to above. the analyses of the systems for the determination of Hg(II) with PAN and Cd(II) and Pb(II) with both extractants were carried out. In the Appendix A values of the predicted desirability functions. Pareto and ANOVA analyses of all metals with both extractants are reported. A comparison of these data with the other processing methods (M2 and M3) will be presented in Section 3.4.

### 3.2. Use of the Absorbances of the Free Chromophore and the Formed Complex (M2)

Searching for an appropriate response to optimize, the maximum values of the absorption bands were tested as a response in the experimental design considering the wavelengths at which the free chromophore band appears (PAN 465 nm, Dithizone 436 nm, and 615 nm) as well as for the metal complex (PAN: Hg 556 and 602 nm, Cd 550 nm, Pb 556 nm; Dithizone: Hg 490 nm, Cd 490 nm, Pb 502 nm). To exemplify this case, the optimization of the membranes prepared using PAN as a chromophore and THEP as a plasticizer for the determination of Pb(II) ions is presented below (Figure 5). 

As clearly observed, the selected bands vary along the experimental design. However. band shifting and distortion are noticeable as well (Figure 5A,B). Once again. the two responses where combined into their desirability function, maximizing the absorption band of the metal complex and minimizing that of the free chromophore:D = 0.535822 + 0.190557 × PAN − 0.0859353 × Time × PAN (6)

The significant term (95% confidence interval), PAN, was chosen according to the Pareto (Figure 6) and ANOVA (Table 4) analyses. With the response surface of the desirability function, an optimal membrane was then selected (Table 5).

The positive influence of PAN indicates that an increase in the amount of this factor maximizes the response; therefore, the optimal values of this factor correspond to the highest level of the chromophore content. However, as its interaction with time is present, PAN contents cannot be analyzed independently of time, denoting a strong kinetic effect for metal extraction.

Similar to above, the analyses of the systems for the determination of Pb(II) with Dz and Hg(II) and Cd(II) with both extractants were carried out. In the Appendix A values of the predicted desirability functions, Pareto and ANOVA analyses and optimal compositions of all metals with both extractants are reported. A comparison of these data with the other preprocessing methods (M1 and M3) will be presented in Section 3.4.

### 3.3. Subtraction of the Normalized Spectra before and after Complexation before PCA Analysis (M3) 

Results of this method are exemplified in the case of the system Cd(II)-PAN. In Figure 7, raw spectral data are shown with their corresponding optode colors. In Figure 8, the results from the spectral subtraction clearly point out that the analysis will be focused in the part of the spectra where the changes are maximal. 

The PCA reduction led to score values in PC-1 and PC-2 that explain 97% of the variability in the data (Figure 9). Once the PCA reduction was applied (Figure 10), the desirability function was computed, maximizing both responses. Its analysis allowed identification of the significant factors (Pareto chart, Figure 11).

This time the loadings plot (Figure 10) practically showed the same profile as that of the subtracted data with the advantage that the chemical meaning of the data reduction technique is not hidden by the abstract meaning of the principal components. The two responses (PC-1 and PC-2) were finally combined into their desirability function, maximizing the individual desirability:D = 0.45101 + 0.105882 × PAN − 0.19029 × Time (7)

The significant terms (95% confidence interval), PAN and Time × PAN, were chosen according to the Pareto (Figure 11) and ANOVA (Table 6) analyses. With the response surface of the desirability function, an optimal membrane was then selected (Table 7).

As PAN content had a positive influence, it means that as the amount of PAN increases, the response increases as well; however, this fact is conditioned by the level of the time variable, with a negative behavior, indicating that time and chromophore content are inversely related.

Similar to above, the analyses of the systems for the determination of Cd(II) with Dz and Hg(II) and Cd(II) with both extractants were carried out. In the Appendix A values of the predicted desirability functions, Pareto and ANOVA analyses of all metals with both extractants are reported. A comparison of these data with the other processing methods (M1 and M2) will be presented in Section 3.4. In Table 7, the found optimal parameters are shown.

### 3.4. Comparison of the M1, M2, and M3 Processing Methods

Two full-spectrum methods (M1 and M3) and one band-based method (M2) were employed. To make an easy comparison along the data, taking advantage of the desirability functions with the same metrics independent of the scale and range of the response variable, coded values of the predictive variables were employed, and a heatmap of the coefficients of all primary and binary significant terms in the desirability functions generated by the different processing methods (Appendix A) was built (Figure 12). In this form, a uniform representation showing the magnitude of the effects of the chromophore, Plasticizer, and CTA content and time are represented in a common color bar ranging from black to yellow according to the extent of their values. It is observed that overall, the process methods tend to give the same importance to the similar variables, although with different weights. Chromophore content and its interaction with time are variables whose importance was expected considering that the color of the optode is determined by the free form of the agent, the complex formed with the metal ion, and the equilibration time. However, the weight (coefficient) of the chromophore follows the sequence M2 < M1 < M3, indicating that the last method is more susceptible to the importance of this variable. This should be a direct consequence of employing wisely all the spectra information using M3, as this method focused on the relevant changes after the complexation of the chromophore and the metal has occurred. In contrast, as discussed previously, M2 is subject to band shifting and distortion. and the PCs in M1 cancel out in some way, which is not easy to comprehend due to the abstract meaning of such figures. The association of CTA content with time and chromophore content is also not strange, as the system functions as a whole. and changes in one variable impacts the others. For the system Dz-Pb^2+^ with M3, the highest coefficient values are observed. Interestingly, the plasticizer content, although it plays its role, is not as pronounced as that of the other variables. This may be a consequence of having selected the appropriate plasticizer for each system from the beginning (THEP for PIMs with PAN and 2NPOE for those with Dz) and using contents at relatively high values. Clearly, each processing method gave characteristic results, but it is necessary to mention that direct comparisons cannot be made as the optimized response varies within each method, so that care should be taken when comparing the signs of the coefficient of the different equations.

To further compare the processing methods, HCA analysis was performed as described in Section 2.4.4. Only one PC was necessary as, on average, it accounts for about 97% of the desirability variations. Dendrograms for the desirability values for each processing method are shown in Figure 13. Although, once again, each method tends to cluster the PIMs in a different form (each experiment in Table 1), going from M1 to M3 the method more precisely discerned the differences in the PIMs, as the nodes among samples appeared at longer distances and varyingly distributed in the dendrogram, i.e., M3 is the method that provides more differentiation amongst PIMs.

It is not evident that the second susceptible method to account for variations in the PIMs is a band-based one (M2), considering that band shifting, and distortion were observed. It seems that although M1 is a full-spectrum method, the abstract meaning of the principal components and its property to model according to the greatest sources of variance hides the differences in the parameters one is interested in capturing.

In conclusion, both the heat map and HCA analyses indicate M3 as the method that gives more discernable differences between the PIMs in the DoE design. In addition, using the naked eye, it was found that M3 produces the higher contrast among PIMs on comparing the uncomplexed versus the complexed membranes. This result is summarized in Table 8.

## 4. Conclusions

The optimization of Hg(II), Cd(II), and Pb(II) optosensors using dithizone and PAN as chromophores to be applied in aqueous solution was carried out successfully by integration of RSM and PCA analyses. The comparison of three different processing methods, i.e., using the score values of the first two components of the PCA data reduction technique of complexed membranes (M1), employing the absorbances of the free chromophore and the formed complex (M2), and using the score values of the first to principal components in PCA after subtraction of the normalized spectra of the membranes before and after complexation (M3), showed that M3 allows detection of more discernable differences between the PIMs in the DoE design, according to both heat map and HCA analyses. Although each processing method gave characteristic results, overall, they tended to give the same importance to similar variables, yet with different weights, according to the ANOVA and Pareto analyses of the coefficients of the PCA reductions of their desirability functions. As M3 focuses on the relevant changes after the complexation of the chromophore and the metal has occurred, the developed full-spectrum method can be used when band-based methods present problems related to overlapping, shifting, and distortion of the signals. In addition, it does not suffer drawbacks associated with the interpretability of full-spectrum methods based only on PCA. Furthermore, HCA clearly showed M3 as the method with more discernable differences between the PIMs in the DoE design, as the nodes among samples appear at longer distances and varyingly distributed in the dendrograms. Due to its easy chemical meaning and the adequate determined color changes, the method is recommended as a novel optimization method for this kind of PIM optode. Applications to multicomponent detection to deconvolute a more complicated system with even more metal ion components are promising areas of future research.

## Figures and Tables

**Figure 1 membranes-11-00288-f001:**
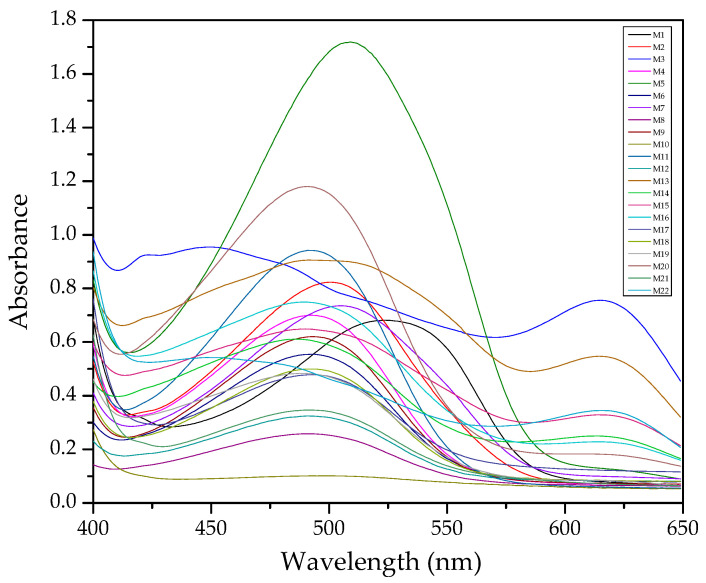
Visible spectra of the membranes after complexation for the system Dz-Hg(II).

**Figure 2 membranes-11-00288-f002:**
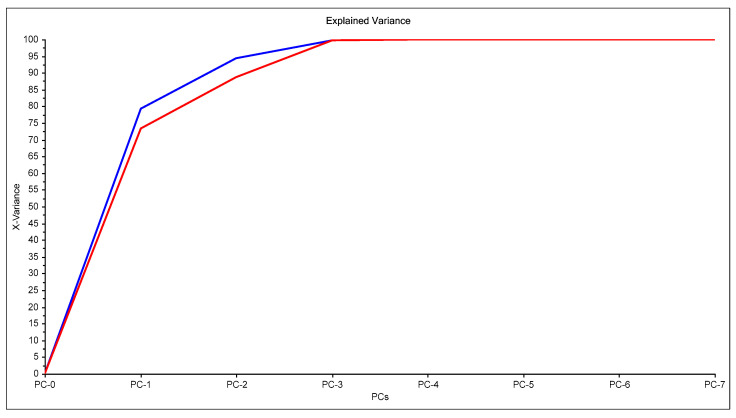
Principal component analysis for the system Dz-Hg(II).

**Figure 3 membranes-11-00288-f003:**
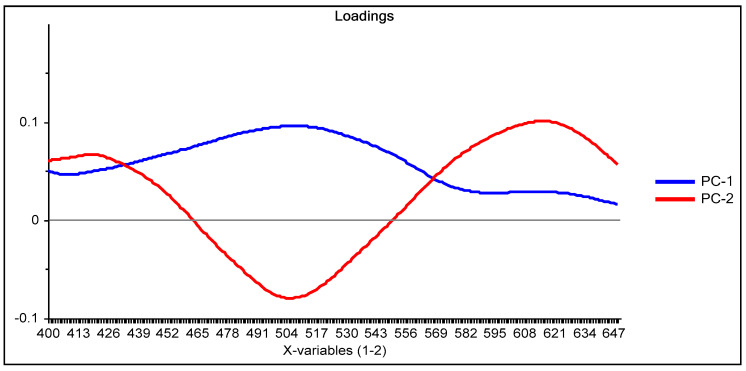
Loading graph of the principal component analysis (PCA) reduction for the system Dz-Hg(II) using the M1 process method of analysis.

**Figure 4 membranes-11-00288-f004:**
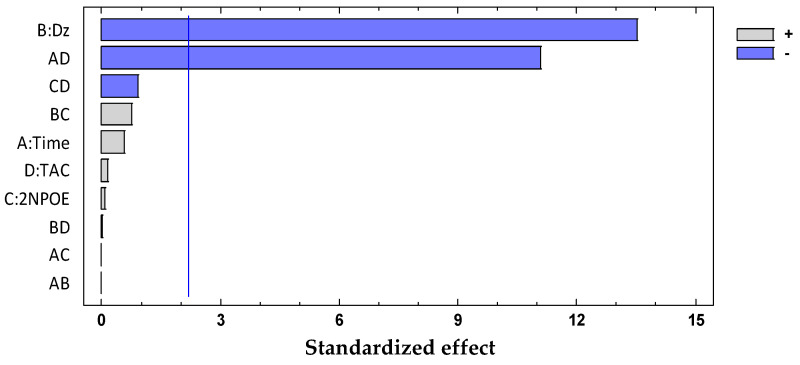
Desirability pareto chart after DoE analysis for the system Dz-Hg(II).

**Figure 5 membranes-11-00288-f005:**
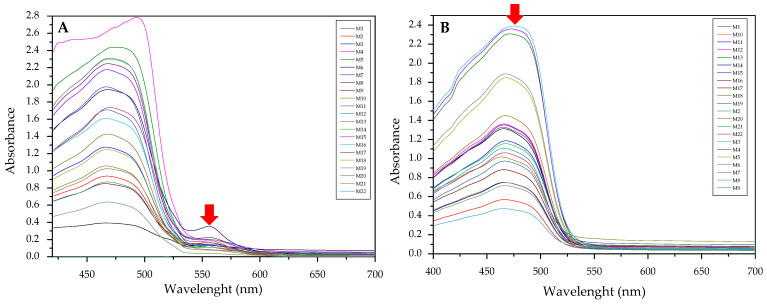
Visible spectra of the membranes before (**A**) and after complexation (**B**) for the system PAN–Pb(II).

**Figure 6 membranes-11-00288-f006:**
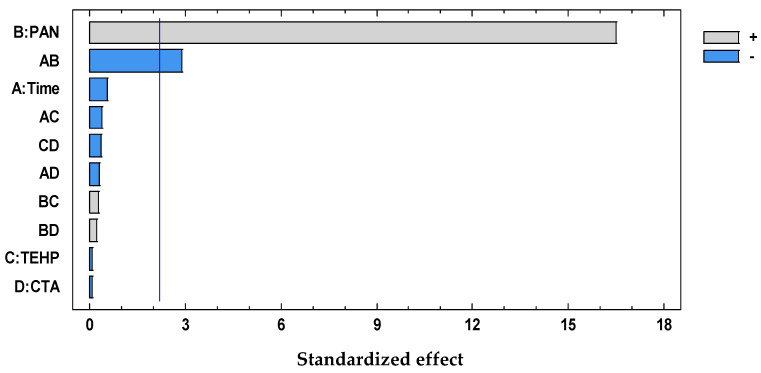
Desirability pareto chart after DoE analysis for the system PAN–Pb(II).

**Figure 7 membranes-11-00288-f007:**
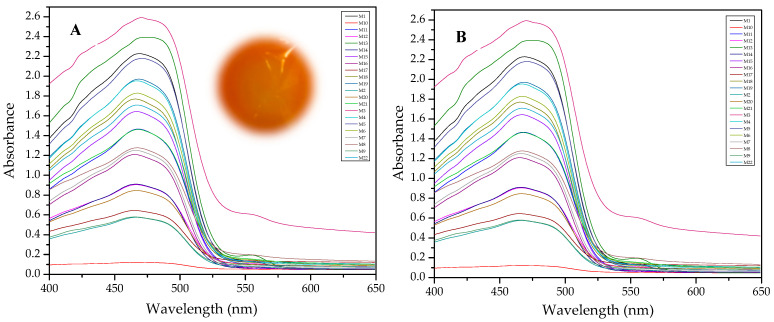
Visible spectra and appearance of the membranes before (**A**) and after (**B**) complexation for the system PAN–Cd(II).

**Figure 8 membranes-11-00288-f008:**
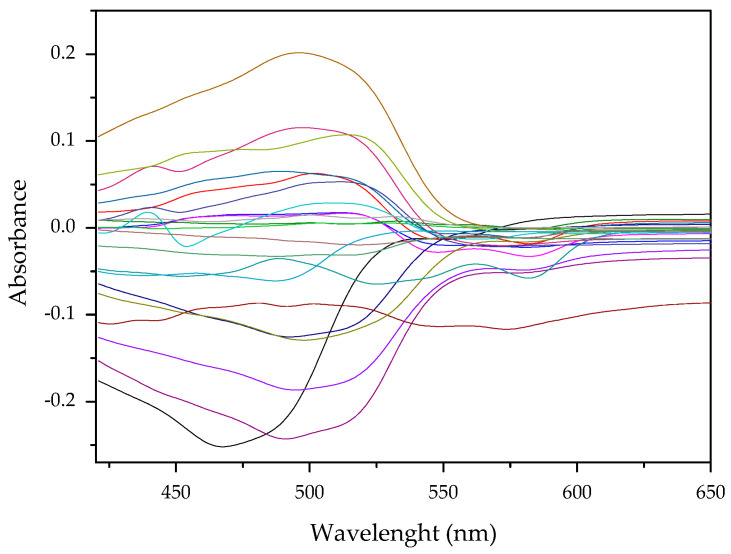
Visible spectra after algebraic transformation for the system PAN–Cd(II).

**Figure 9 membranes-11-00288-f009:**
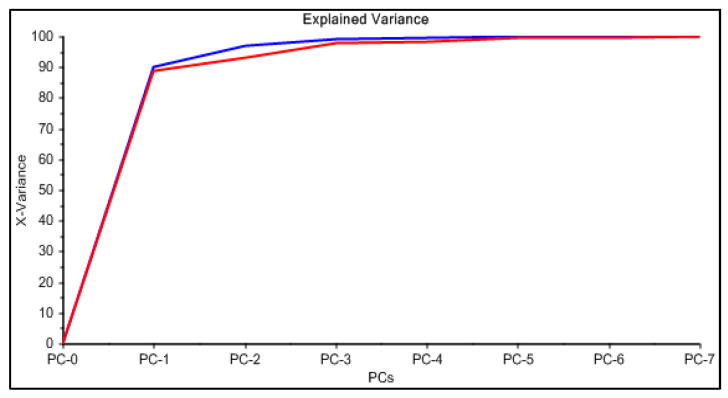
Principal component analysis for the system PAN–Cd(II).

**Figure 10 membranes-11-00288-f010:**
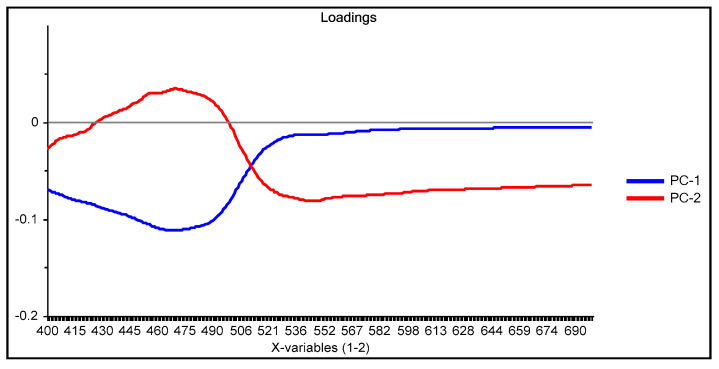
Loadings plot for the M3 processing method in the case of the PAN-Cd(II) PIM optode.

**Figure 11 membranes-11-00288-f011:**
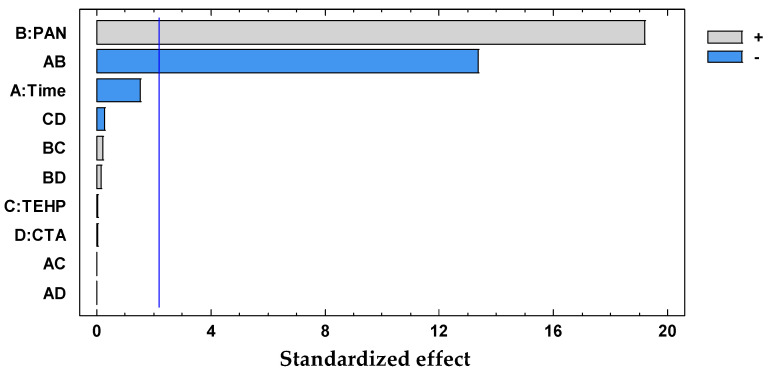
Desirability pareto chart after DoE analysis for the system PAN–Cd(II).

**Figure 12 membranes-11-00288-f012:**
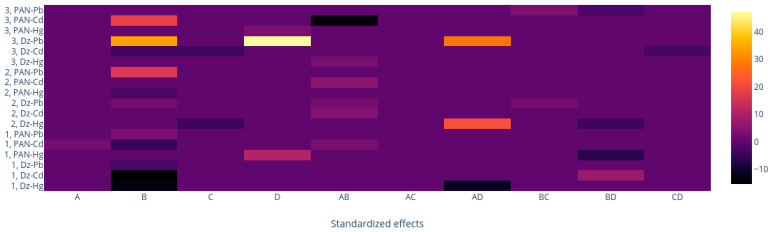
Heatmap of the coefficients of the desirability functions generated by the three different processing methods (M1. M2. M3). A: Time; B: Chromophore. C: Plasticizer; D: CTA.

**Figure 13 membranes-11-00288-f013:**
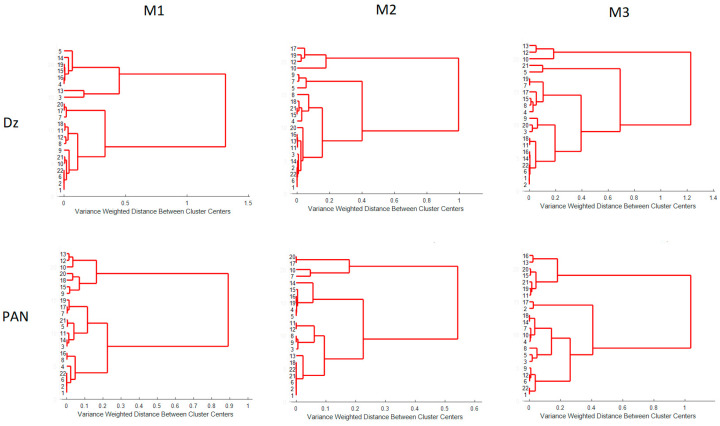
Dendrograms for the desirability values for M1, M2, and M3.

**Table 1 membranes-11-00288-t001:** Doehlert matrix employed to study the influence of the membrane composition and the equilibration time on the performance of the optodes. Coded and real values of the variables are indicated.

Experimental Runs	Factor A	Factor B	Factor C	Factor D
	Time	Chromophore	Plasticizer	CTA
1	0 (50 min)	0 (0.53 mg)	0 (62.5 mg)	0 (62.5 mg)
2	1 (80 min)	0 (0.53 mg)	0 (62.5 mg)	0 (62.5 mg)
3	0.5 (65 min)	0.866 (1 mg)	0 (62.5 mg)	0 (62.5 mg)
4	0.5 (65 min)	0.289 (0.68 mg)	0.817 (100 mg)	0 (62.5 mg)
5	0.5 (65 min)	0.289 (0.68 mg)	0.204 (71.86 mg)	0.791 (100 mg)
6	−1 (20 min)	0 (0.53 mg)	0 (62.5 mg)	0 (62.5 mg)
7	−0.5 (35 min)	−0.866 (0.060 mg)	0 (62.5 mg)	0 (62.5 mg)
8	−0.5 (35 min)	−0.289 (0.37 mg)	−0.817 (25.0 mg)	0 (62.5 mg)
9	−0.5 (35 min)	−0.289 (0.37 mg)	−0.204 (53.13 mg)	−0.791 (25 mg)
10	0.5 (65 min)	−0.866 (0.06 mg)	0 (62.5 mg)	0 (62.5 mg)
11	0.5 (65 min)	−0.289 (0.37 mg)	−0.817 (25 mg)	0 (62.5 mg)
12	0.5 (65 min)	−0.289 (0.37 mg)	−0.204 (53.13 mg)	−0.791 (25 mg)
13	−0.5 (35 min)	0.866 (1.0 mg)	0 (62.5 mg)	0 (62.5 mg)
14	0 (50 min)	0.577 (0.84 mg)	−0.817 (25 mg)	0 (62.5 mg)
15	0 (50 min)	0.577 (0.84 mg)	−0.204 (53.13 mg)	−0.791 (25 mg)
16	−0.5 (35 min)	0.289 (0.68 mg)	0.817 (100 mg)	0 (62.5 mg)
17	0 (50 min)	−0.577 (0.21 mg)	0.817 (100 mg)	0 (62.5 mg)
18	0 (50 min)	0 (0.53 mg)	0.613 (90.63 mg)	−0.791 (25 mg)
19	−0.5 (35 min)	0.289 (0.68 mg)	0.204 (71.86 mg)	0.791 (100 mg)
20	0 (50 min)	−0.577 (0.21 mg)	0.204 (71.86 mg)	0.791 (100 mg)
21	0 (50 min)	0 (0.53 mg)	−0.613 (34.36 mg)	0.791 (100 mg)

**Table 2 membranes-11-00288-t002:** Analysis of variance (ANOVA) results for the system Dz-Hg(II).

Source	Sum of Squares	Df	Mean Square	F-Ratio	*p*-Value
**A:Time**	0.0000836037	1	0.0000836037	0.33	0.5759
**B:Dz**	0.0460006	1	0.0460006	182.85	**0.0000**
**C:2NPOE**	0.0000018768	1	0.0000018768	0.01	0.9327
**D:CTA**	0.00000643161	1	0.00000643161	0.03	0.8759
**AB**	0.0	1	0.0	0.00	1.0000
**AC**	0.0	1	0.0	0.00	1.0000
**AD**	0.0309215	1	0.0309215	122.91	**0.0000**
**BC**	0.0001458	1	0.0001458	0.58	0.4625
**BD**	1.26293 × 10^−7^	1	1.26293 × 10^−7^	0.00	0.9825
**CD**	0.000213563	1	0.000213563	0.85	0.3766
**Total Error**	0.00276736	11	0.000251578		
**Total (corrected)**	0.0840181	21			
**R^2^**	96.7062%				
**Adj − R^2^**	93.7119%				
**Standard error**	0.0158612				
**Std. Dev**	0.00881342				

**Table 3 membranes-11-00288-t003:** Summary of the optimization results for the membranes prepared with PAN and THEP using M1 in the determination of Hg(II). Pb(II). and Cd(II). Coded and real values of the variables are indicated.

Metal	Optimal Composition			
	Time	PAN	THEP	CTA
**Hg^2+^**	0.5 (65 min)	−0.289 (0.37 mg)	−0.204 (53.13 mg)	−0.791 (25 mg)
**Pb^2+^**	0 (50 min)	−0.577 (0.21 mg)	0.204 (71.86 mg)	0.791 (100 mg)
**Cd^2+^**	−0.5 (35 min)	−0.866 (0.600 mg)	0 (62.5 mg)	0 (62.5 mg)

**Table 4 membranes-11-00288-t004:** Analysis of variance (ANOVA) results for the system PAN-Pb(II).

Source	Sum of Squares	Df	Mean Square	F-Ratio	*p*-Value
**A:Time**	0.000200484	1	0.000200484	0.30	0.5941
**B:PAN**	0.181549	1	0.181549	272.70	**0.0000**
**C:THEP**	0.00000417595	1	0.00000417595	0.01	0.9383
**D:CTA**	0.00000250761	1	0.00000250761	0.00	0.9522
**AB**	0.00553834	1	0.00553834	8.32	**0.0149**
**AC**	0.000100039	1	0.000100039	0.15	0.7057
**AD**	0.0000556332	1	0.0000556332	0.08	0.7779
**BC**	0.0000519484	1	0.0000519484	0.08	0.7852
**BD**	0.0000275653	1	0.0000275653	0.04	0.8425
**CD**	0.0000784508	1	0.0000784508	0.12	0.7379
**Total Error**	0.00732311	11	0.000665738		
**Total (corrected)**	0.197303	21			
**R^2^**	96.2884%				
**Adj − R^2^**	92.9142%				
**Standard error**	0.0258019				
**Std. Dev**	0.0135246				

**Table 5 membranes-11-00288-t005:** Summary of the optimization results for the membranes prepared with PAN using M2 in the determination of Hg(II), Pb(II), and Cd(II).

Metal	Wavelength (nm)		Optimal Composition *
	Free Chromophore (PAN)	Metal Complex	
Hg^2+^	465	556	8
Pb^2+^	465	556	13
Cd^2+^	465	550	14

* Refers to the number of experimental runs in Table 1.

**Table 6 membranes-11-00288-t006:** Analysis of variance (ANOVA) results for the system PAN-Cd(II).

Source	Sum of Squares	Df	Mean Square	F-Ratio	*p*-Value
**A:Time**	0.00035041	1	0.00035041	2.30	0.1573
**B:PAN**	0.0560517	1	0.0560517	368.35	**0.0000**
**C:THEP**	2.38862 × 10^−7^	1	2.38862 × 10^−7^	0.00	0.9691
**D:CTA**	1.43434 × 10^−7^	1	1.43434 × 10^−7^	0.00	0.9761
**AB**	0.0271562	1	0.0271562	178.46	**0.0000**
**AC**	3.5837 × 10^−8^	1	3.5837 × 10^−8^	0.00	0.9880
**AD**	1.99296 × 10^−8^	1	1.99296 × 10^−8^	0.00	0.9911
**BC**	0.00000700214	1	0.00000700214	0.05	0.8341
**BD**	0.00000371548	1	0.00000371548	0.02	0.8787
**CD**	0.0000103725	1	0.0000103725	0.07	0.7989
**Total Error**	0.00167386	11	0.000152169		
**Total (corrected)**	0.0913495	21			
**R^2^**	98.1676%				
**Adj − R^2^**	96.5018%				
**Standard error**	0.012357				
**Std. Dev**	0.00509949				

**Table 7 membranes-11-00288-t007:** Optimal parameters obtained after applying the M3 processing method.

Metal	Optimal Composition							
	Time	Dithizone	NPOE	CTA	Time	PAN	THEP	CTA
**Hg^2+^**	65 min	0.68 mg	71.86 mg	100 mg	50 min	0.53 mg	34.36 mg	100 mg
**Cd^2+^**	35 min	0.60 mg	62.5 mg	62.5 mg	35 min	1.0 mg	62.5 mg	62.5 mg
**Pb^2+^**	65 min	0.68 mg	71.86 mg	100 mg	35 min	0.6 mg	62.5 mg	62.5 mg

**Table 8 membranes-11-00288-t008:** Results for optimization using the PIM optodes and the M3 process method.

System	Optimal Experiment	Appearance of the Membrane		Spectra
		Before	After	
**PAN + Hg**	21	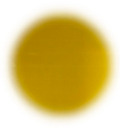	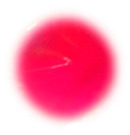	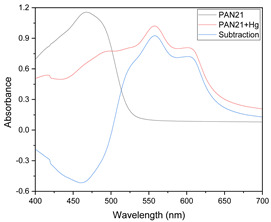
**PAN + Cd**	13	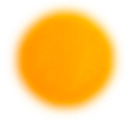	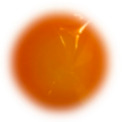	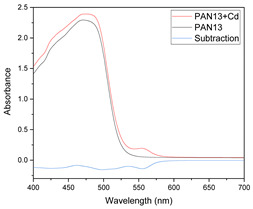
**PAN + Pb**	7	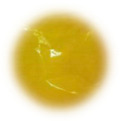	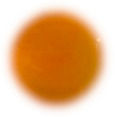	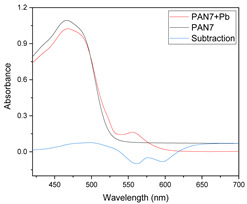
**Dz + Hg**	5	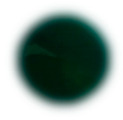	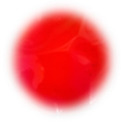	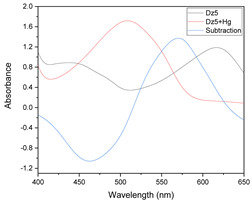
**Dz + Cd**	7	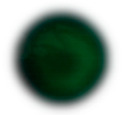	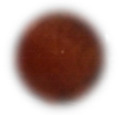	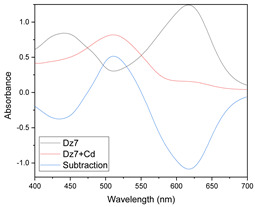
**Dz + Pb**	5	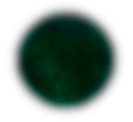	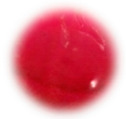	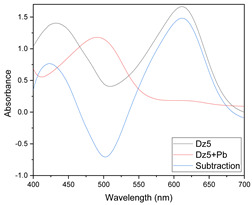

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
