# Peer review of "Integration of Response Surface Methodology (RSM) and Principal Component Analysis (PCA) as an Optimization Tool for Polymer Inclusion Membrane Based-Optodes Designed for Hg(II), Cd(II), and Pb(II)"

_membranes, 2021, doi:10.3390/membranes11040288_

Round 1

Reviewer 1 Report

The manuscript entitled “Integration of response surface methodology (RSM) and principal component analysis (PCA) as an optimization tool for polymer inclusion membrane based-optodes designed for Hg(II), Cd(II), and Pb(II).” by García-Beleño and San Miguel, is an interesting topic in the field of environment engineering, but for further proceeding of this paper, there are some major concerns, which decision on this paper, can be taken after evaluating the provided responses by Authors.

Comments:

  1. Abstract is too long, it should be revised by reporting more numerical values and results.
  2. Introduction should be revised by referring and presenting more recent and relevant studies, the following studies can be helpful ones:
  • Shirzad et al., Desalination and Water Treatment.144 (2019) 185-200.
  • Kalla, Journal of Environmental Chemical Engineering, 9 (2021) 104641
  • Wang et al., Chemical Engineering Science, 220, (2020), 115620.
  • ……
  1. It is necessary, one subsection to be added to the Introduction section, by referring more discussion and explanation about Response Surface Methodology (RSM) topology. Following references can be interesting ones:
  • Henrique et al., Chemical Engineering & Technology, 42 (2019) 327-342.
  • Shirzad et al., Desalination and Water Treatment, 182 (2020) 194-207.
  • H. Myers, D.C. Montgomery, C.M. Anderson-Cook, Response surface methodology: process and product optimization using designed experiments, 4th ed., Wiley Publication, New Jersey, 2016.
  • Karimi et al., Industrial & Engineering Chemistry Research, 57 (2018) 11154-11166.
  • ……
  1. Equations 2, 3 and 4 are not legible, they should be presented in a high quality format.
  2. The paper suffers from poor Language and strongly is recommended to be checked and revised by a native speaker.

Results & Discussion:

  1. The information in Figure 1 (a) and (b) is not legible; the figures should be presented in high quality formats.
  2. The following information should be presented in ANOVA results:

Coded coefficient, Standard error, Lack-of-fit, Residual, Std. dev, R2, Adj-R2

  1. It is strongly recommend presenting the results and discussion based on obtained “Coded Coefficients” values. More details can be found in:
  • Shirzad et al., Desalination and Water Treatment, 182 (2020) 194-207.
  1. To have a better grasp on the considered factors and developed surfaces, 3D-plots of derived models should be presented and discussed.
  2. As can be observed in derived polynomial equations (Eqs5-7 and in SM), the order of some of factors are really lower than other ones, which means, these terms can be ignored in the developed models, thus explanations should be presented based on remaining factors.
  3. Figures 3 and 4 are not legible; they should be presented in high quality formats.
  4. More scientific explanation is required for developed “Heat maps”
  5. The conclusion should be revised by reporting more numerical results.
  6. The explanation in Supplementary Materials (SM) section should be completely moved to Supplementary Materials!!! In the manuscript, just a summery (around 2-3 sentences) is required.

Author Response

The manuscript entitled “Integration of response surface methodology (RSM) and principal component analysis (PCA) as an optimization tool for polymer inclusion membrane based-optodes designed for Hg(II), Cd(II), and Pb(II).” by García-Beleño and San Miguel, is an interesting topic in the field of environment engineering, but for further proceeding of this paper, there are some major concerns, which decision on this paper, can be taken after evaluating the provided responses by Authors.

Comments:

  1. Abstract is too long, it should be revised by reporting more numerical values and results.

R: Due to the qualitative nature of the work, aimed to obtain the best composition of PIM optosensors by application of DoE and PCA, numerical values reflecting the obtained results can not be directly extracted. Table 8 summarized the results. To be more descriptive about experimental results, the abstract was rewritten, as suggested.

  1. Introduction should be revised by referring and presenting more recent and relevant studies, the following studies can be helpful ones:
  • Shirzad et al., Desalination and Water Treatment.144 (2019) 185-200.
  • Kalla, Journal of Environmental Chemical Engineering, 9 (2021) 104641
  • Wang et al., Chemical Engineering Science, 220, (2020), 115620.
  • ……

R: Done (line 75 revised version).

  1. It is necessary, one subsection to be added to the Introduction section, by referring more discussion and explanation about Response Surface Methodology (RSM) topology. Following references can be interesting ones:
  • Henrique et al., Chemical Engineering & Technology, 42 (2019) 327-342.
  • Shirzad et al., Desalination and Water Treatment, 182 (2020) 194-207.
  • Myers, D.C. Montgomery, C.M. Anderson-Cook, Response surface methodology: process and product optimization using designed experiments, 4th ed., Wiley Publication, New Jersey, 2016.
  • Karimi et al., Industrial & Engineering Chemistry Research, 57 (2018) 11154-11166.
  • ……
  1. Done. Section 2.4.1 was added.
  1. Equations 2, 3 and 4 are not legible, they should be presented in a high quality format.

R: Done.

  1. The paper suffers from poor Language and strongly is recommended to be checked and revised by a native speaker.

R: Done, the entire manuscript was revised in this aspect.

Results & Discussion:

  1. The information in Figure 1 (a) and (b) is not legible; the figures should be presented in high quality formats.

R: Done. The Figures were separated and renumbered.

  1. The following information should be presented in ANOVA results:

Coded coefficient, Standard error, Lack-of-fit, Residual, Std. dev, R2, Adj-R2

                R: Done. Parity plots were not included as the supplementary information contains now to much information. All equations with coded coefficients considering only significant terms are given in the SI.

  1. It is strongly recommend presenting the results and discussion based on obtained “Coded Coefficients” values. More details can be found in:
  • Shirzad et al., Desalination and Water Treatment, 182 (2020) 194-207.

R: All analyses were performed with coded values. This is now precise in the manuscript (lines 105-107 revised version).

  1. To have a better grasp on the considered factors and developed surfaces, 3D-plots of derived models should be presented and discussed.

R: Done, see SI.

  1. As can be observed in derived polynomial equations (Eqs5-7 and in SM), the order of some of factors are really lower than other ones, which means, these terms can be ignored in the developed models, thus explanations should be presented based on remaining factors.

R: Done, the heatmap was also re-drawn considering only significant factors.

  1. Figures 3 and 4 are not legible; they should be presented in high quality formats.

R: Done.

  1. More scientific explanation is required for developed “Heat maps”

R: Done. Additional explanations were included in lines 533-535, 546-550 of the revised version.

  1. The conclusion should be revised by reporting more numerical results.

As previously indicated, the nature of the work is not quantitative. The conclusions were rewritten to emphasize the obtained results.

  1. The explanation in Supplementary Materials (SM) section should be completely moved to Supplementary Materials!!! In the manuscript, just a summery (around 2-3 sentences) is required.

R: Done.

Reviewer 2 Report

The manuscript present a optimization tool for poly-3 mer inclusion membrane based-optodes  designed for Hg(II), 4 Cd(II), and Pb(II). The work is interesting but the presentation is confuse . the reference lis is complete

Specific commnets:

  • What is the influence of temperature and pH?
  • Explain the equation 1.
  • Check the writing of equations.
  • Table 4 is confuse.
  •  Figure 6 is confuse
  • What is the better solution for each parameter of detection?
  • Is possible use this membranes in optical system? Applications?
  • What is the response of membrane to the different metal
  • How discriminate each metal?

I recommned the authors rewrite the manuscript and present a new version. I suggest a major review.

Author Response

The manuscript present a optimization tool for poly-3 mer inclusion membrane based-optodes  designed for Hg(II), 4 Cd(II), and Pb(II). The work is interesting but the presentation is confuse . the reference lis is complete

 Along the work, the optimizations of four PIM optosensors variables (CTA, plasticizer, chromophore, and time) using Doehlert experimental design matrices for three metals and two chromophores were performed. Three different processing methods for the response variable, two of them based on PCA reduction, were applied and compared. To simplify the discussion, only representative cases of the analyses were presented in the main manuscript and the complete analyses were shown in the supplementary information. The comparison of the three processing methods of the response was performed using heatmap and cluster analyses. The description of the utility and meaning of RSM, Desirability functions, PCA, HCA and heat maps were given after the introduction. We have made our best effort to maintain the presentation as clear as possible. We hope that the changes made in the revised manuscript will help to fully understand the used approach.

Specific commnets:

  • What is the influence of temperature and pH?

These variables were not studied at this time and further work will be performed in this aspect when the developed optosensors will be applied.

  • Explain the equation 1.
  • R: An explanation was included in lines 173-178 of the revised version.
  • Check the writing of equations.
  • R: Done.
  • Table 4 is confuse.

R: It was clarify stating the final compositions.

  • Figure 6 is confuse
  • R: Sentences were added to clarify the Figure in lines 537-540 of the revised version.
  • What is the better solution for each parameter of detection?

The final optimal experimental conditions are given in Table 8 referring to the experimental runs reported in Table 1.

  • Is possible use this membranes in optical system? Applications?
  • R: Applications of PIM optosensors were already given in the introduction section.
  • What is the response of membrane to the different metal

R: The response varied depending on the processing method, i.e., is not the same for M1, M2 or M3, as discussed along the work. M1 and M3 responses are transformations of the full absorbance spectrum of the optosensors while M2 is based on the absorbances of two characteristics bands (chomophore and chromophore-metal complex). To evaluate them simultaneously, desirability functions were employed.

  • How discriminate each metal?

R: Now only mono-elemental solutions were employed. Application to more complex systems will be addressed in future research as pointed out in the conclusions.

I recommned the authors rewrite the manuscript and present a new version. I suggest a major review.

Reviewer 3 Report

The authors reported the optimization of Hg (II), Cd (II), and Pb (II) optosensors using a combination of response surface methodology and principal component analysis. The paper is well written, and the integrated approach proposed by this work is quite interesting. I believe it is suitable for publication in Membranes.

Here are a few thoughts:

  • Would the integrated approach still be effective to deconvolute a more complicated system with even more metal ion components (i.e., potentially mixed colors)?
  • Would it give sufficient information about a system that generates relatively non-uniform colors (non-homogeneous systems)?
  • It would be good to include scale bars for those optical images that show the appearances of the membranes.
  • The axis labels in Figure 1A and 1B are too small to see.

Author Response

The authors reported the optimization of Hg (II), Cd (II), and Pb (II) optosensors using a combination of response surface methodology and principal component analysis. The paper is well written, and the integrated approach proposed by this work is quite interesting. I believe it is suitable for publication in Membranes.

Here are a few thoughts:

  • Would the integrated approach still be effective to deconvolute a more complicated system with even more metal ion components (i.e., potentially mixed colors)?

R: This is a very interesting point to be treated in future research. This idea was included in the conclusions.

  • Would it give sufficient information about a system that generates relatively non-uniform colors (non-homogeneous systems)?

Membranes were maintained homogeneous through the addiction of ethanol. This is now emphasized in the manuscript in lines 133-134 of the revised manuscript.

  • It would be good to include scale bars for those optical images that show the appearances of the membranes.

R: the supplementary information now includes all the studied optosensors to have a better insight of the results.

  • The axis labels in Figure 1A and 1B are too small to see.

R: Done

Round 2

Reviewer 1 Report

As already mentioned, the manuscript entitled “Integration of response surface methodology (RSM) and principal component analysis (PCA) as an optimization tool for polymer inclusion membrane based-optodes designed for Hg(II), Cd(II), and Pb(II).” by García-Beleño and San Miguel, is an interesting topic in the field of environment engineering.

The Authors correctly replied to the requested editions/comments, thus the paper can be accepted in the present format.

Reviewer 2 Report

the authors present a reviewed version whre clarify the previous questions. I recommend publish